# Characterizing Structural Regularities of Labeled Data in Overparameterized Models

## Abstract

Humans are accustomed to environments that contain both regularities and exceptions. For example, at most gas stations, one pays prior to pumping, but the occasional rural station does not accept payment in advance. Likewise, deep neural networks can generalize across instances that share common patterns or structures, yet have the capacity to memorize rare or irregular forms. We analyze how individual instances are treated by a model via a *consistency score*. The score characterizes the expected accuracy for a held-out instance given training sets of varying size sampled from the data distribution. We obtain empirical estimates of this score for individual instances in multiple data sets, and we show that the score identifies out-of-distribution and mislabeled examples at one end of the continuum and strongly regular examples at the other end. We identify computationally inexpensive proxies to the consistency score using statistics collected during training. We apply the score toward understanding the dynamics of representation learning and to filter outliers during training.

## 1 Introduction

Human learning requires both inferring regular patterns that generalize across many distinct examples and memorizing irregular examples. The boundary between regular and irregular examples can be fuzzy. For example, in learning the past tense form of English verbs, there are some verbs whose past tenses must simply be memorized (GO→WENT, EAT→ATE, HIT→HIT) and there are many *regular* verbs that obey the rule of appending "ed" (KISS→KISSED, KICK→KICKED, BREW→BREWED, etc.). Generalization to a novel word typically follows the "ed" rule, for example, BINK→BINKED. Intermediate between the exception verbs and regular verbs are subregularities—a set of exception verbs that have consistent structure (e.g., the mapping of SING→SANG, RING→RANG). Note that rule-governed and exception cases can have very similar forms, which increases the difficulty of learning each. Consider one-syllable verbs containing 'ee', which include the regular cases NEED→NEEDED as well as exception cases like SEEK→SOUGHT. Generalization from the rule-governed cases can hamper the learning of the exception cases and vice-versa. For instance, children in an environment where English is spoken over-regularize by mapping GO→GOED early in the course of language learning. Neural nets show the same interesting pattern for verbs over the course of training (Rumelhart & McClelland, 1986).

Memorizing irregular examples is tantamount to building a look-up table with the individual facts accessible for retrieval. Generalization requires the inference of statistical regularities in the training environment, and the application of procedures or rules for exploiting the regularities. In deep learning, memorization is often considered a failure of a network because memorization implies no generalization. However, mastering a domain involves knowing when to generalize and when not to generalize, because the data manifolds are rarely unimodal. Consider the two-class problem of chair vs non-chair with training examples illustrated in Figure 1a. The iron throne (lower left) forms a sparsely populated mode (*sparse mode* for short) as there may not exist many similar cases in the data environment. Generic chairs (lower right) lie in a region with a consistent labeling (a densely populated mode, or *dense mode*) and thus seems to follow a strong regularity. But there are many other cases in the continuum of the two extreme. For example, the rocking chair (upper right) has a few supporting neighbors but it lies in a distinct neighborhood from the majority of same-label instances (the generic chairs).

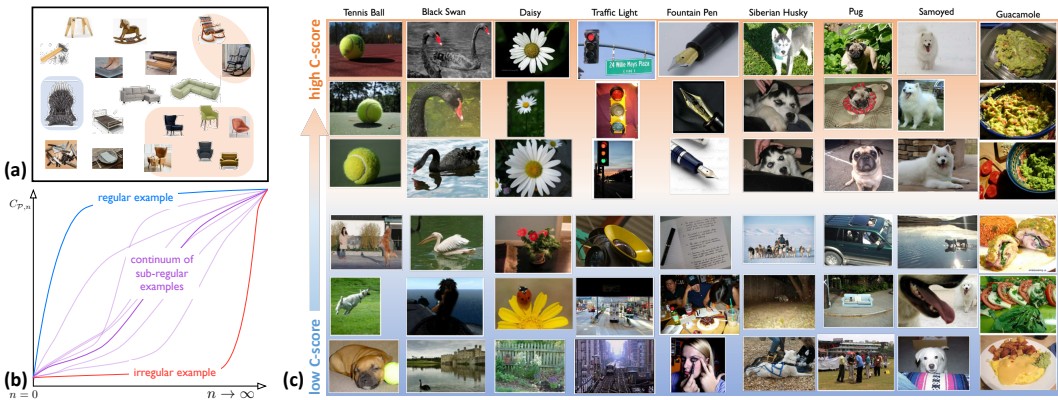

Figure 1: Regularities and exceptions in a binary chairs vs non-chairs problem. (b) illustration of consistency profiles. (c) Regularities (high C-scores) and exceptions (low C-scores) in ImageNet.

In this article, we study this continuum of the structural regularities of data sets in the context of training overparameterized deep networks. Let $D \overset{n}{\sim} \mathcal{P}$ be an i.i.d. sample of size $n$ from the underlying data distribution $\mathcal{P}$, and $f(\cdot\,; D)$ be a model trained on $D$. For an instance $x$ with label $y$, we trace out the following *consistency profile* by increasing $n$:

$$C_{\mathcal{P},n}(x,y) = \mathbb{E}_{D \overset{n}{\sim} \mathcal{P}}[\mathbb{P}(f(x; D \backslash \{(x,y)\}) = y], \quad n = 0, 1, \ldots \tag{1}$$

This quantity measures the per-instance generalization on $(x, y)$. For a fixed $n$, it characterizes how consistent $(x, y)$ is with a sample $D$ from $\mathcal{P}$. Formally, it is closely tied to the fundamental notion of *generalization performance* when we take an expectation over $(x, y)$. Without the expectation, the quantity gives us a fine-grain characterization of the regularity of each individual example. This article focuses on classification problems, but the definition can be easily extended to other problems by replacing the 0-1 classification loss with another suitable loss function.

The quantity $C_{\mathcal{P},n}(x, y)$ has an interpretation that matches our high-level intuition about the structural regularities of the training data during (human or machine) learning. In particular, we can characterize the multimodal structure of an underlying data distribution by grouping examples in terms of a model's generalization profile for those examples when trained on data sets of increasing size. For $n = 0$, the model makes predictions entirely based on its prior belief. As $n$ increases, the model collects more information about $\mathcal{P}$ and makes better predictions. For an $(x, y)$ instance belonging to a dense mode (e.g., the generic chairs in Figure 1a), the model prediction is accurate even for small $n$ because even small samples have many class-consistent neighbors. The blue curve in the cartoon sketch of Figure 1b illustrates this profile. For instances belonging to sparse modes (e.g., the iron throne in Figure 1a), the prediction will be inaccurate for even large $n$, as the red curve illustrates. Most instances fill the continuum between these two extreme cases, as illustrated by the purple curves in Figure 1b. To obtain a *total ordering* for all examples, we pool the consistency profile into a scalar *consistency score*, or *C-score* by taking expectation over $n$. Figure 1c shows examples from the ImageNet data set ranked by estimated C-scores, using a methodology we shortly describe. The images show that on many ImageNet classes, there exist dense modes of center-cropped, close-up shot of the representative examples; and at the other end of the C-score ranking, there exist sparse modes of highly ambiguous examples (in many cases, the object is barely seen or can only be inferred from the context in the picture).

With strong ties to both theoretical notions of learning and human intuition, the consistency profile is an important tool for understanding the regularity and subregularity structures of training data sets and the learning dynamics of models trained on those data. The C-score based ranking also has many potential uses, such as detecting out-of-distribution and mislabeled instances; balancing learning between dense and sparse modes to ensure fairness when learning with data from underrepresented groups; or even as a diagnostic used to determine training priority in a curriculum learning setting (Bengio et al., 2009; Saxena et al., 2019). In this article, we focus on formulating and analyzing consistency profiles, and apply the C-score to analyzing the structure of real world image data sets and the learning dynamics of different optimizers. We also study efficient proxies and further applications to outlier detection.

Our key contributions are as follows:

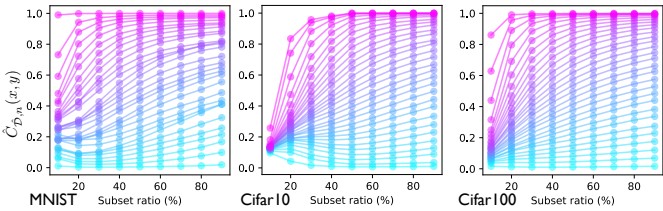

Figure 2: Consistency profiles of training examples. Each curve in the figure corresponds to the average profile of a set of examples, partitioned according to the area under the profile curve of each example.

- We formulate and analyze a consistency score that takes inspiration from generalization theory and matches our high level intuitions.
- We estimate the C-scores with a series of approximations and apply the measure to analyze the structural regularities of the MNIST, CIFAR-10, CIFAR-100, and ImageNet training sets.
- We demonstrate using C-scores to help analyze the learning dynamics of different optimizers and consequences for generalization.
- We evaluate a number of learning-speed based measures as efficient proxies for the C-score, and find that the speed at which an example is learned is a strong indicator of its C-score ranking. While this observation may be intuitive in retrospect, it is nontrivial because learning speed is measured on training examples and the C-score is defined for hold-out generalization. An efficiently computable proxy has practical implications, which we demonstrate with an outlier-detection experiment.
- Because the C-score is generally useful for analyzing data sets and understanding the mechanics of learning, we have released the pre-computed C-scores at (URL anonymized), together with model checkpoints, code, and extra visualizations can also be downloaded.

## 2 RELATED WORK

Analyzing the structure of data sets has been a central topic for many fields like Statistics, Data Mining and Unsupervised Learning. In this paper, we focus on supervised learning and the interplay between the regularity structure of data and overparameterized neural network learners. This differentiates our work from classical analyses based on input or (unsupervised) latent representations. The distinction is especially prominent in deep learning where a supervised learner jointly learns the classifier and the representation that captures the semantic information in the labels.

In the context of deep supervised learning, Carlini et al. (2018) proposed measures for identifying *prototypical* examples which could serve as a proxy for the complete data set and still achieve good performance. These examples are not necessarily the center of a dense neighborhood, which is what our high C-score measures. Two prototype measures explored in Carlini et al. (2018), *model confidence* and the *learning speed*, are also measures we examine. Their *holdout retraining* and *ensemble agreement* metrics are conceptually similar to our C-score estimation algorithm. However, their retraining is a two-stage procedure involving pre-training and fine-tuning; their ensemble agreement mixes architectures with heterogeneous capacities and ignores labels. Feldman (2020) and Feldman & Zhang (2020) studied the positive effects of memorization on generalization by measuring the influence of a training example on a test example, and identifying pairs with strong influences. To quantify memorization, they defined a memorization score for each $(x, y)$ in a training set as the drop in prediction accuracy on $x$ when $(x, y)$ is removed. A point evaluation of our consistency profile on a fixed data size $n$ resembles the second term of their score. A key difference is that we are interested in the profile with increasing $n$, i.e. the sample complexity required to correctly predict $(x, y)$.

We evaluate various cheap-to-compute *proxies* for the C-score and found that the learning speed has a strong correlation with the C-score. Learning speed has been previously studied in contexts quite different from our focus on generalization of individual examples. Mangalam & Prabhu (2019) show that examples learned first are those that could be learned by shallower nets. Hardt et al. (2016) present theoretical results showing that the generalization gap is small if SGD training completes in relatively few steps. Toneva et al. (2019) study forgetting (the complement of learning speed) and informally relate forgetting to examples being outliers or mislabeled. There is a large literature of criteria with no explicit ties to generalization as the C-score has, but provides a means of stratifying instances. For example, Wu et al. (2018) measure the difficulty of an example by the number of residual blocks in a ResNet needed for prediction.

## 3 THE CONSISTENCY PROFILE AND THE C-SCORE

The consistency profile (Equation 1) encodes the structural consistency of an example with the underlying data distribution $\mathcal{P}$ via expected performance of models trained with increasingly large data sets sampled from $\mathcal{P}$. However, it is not possible to directly compute this profile because $\mathcal{P}$ is

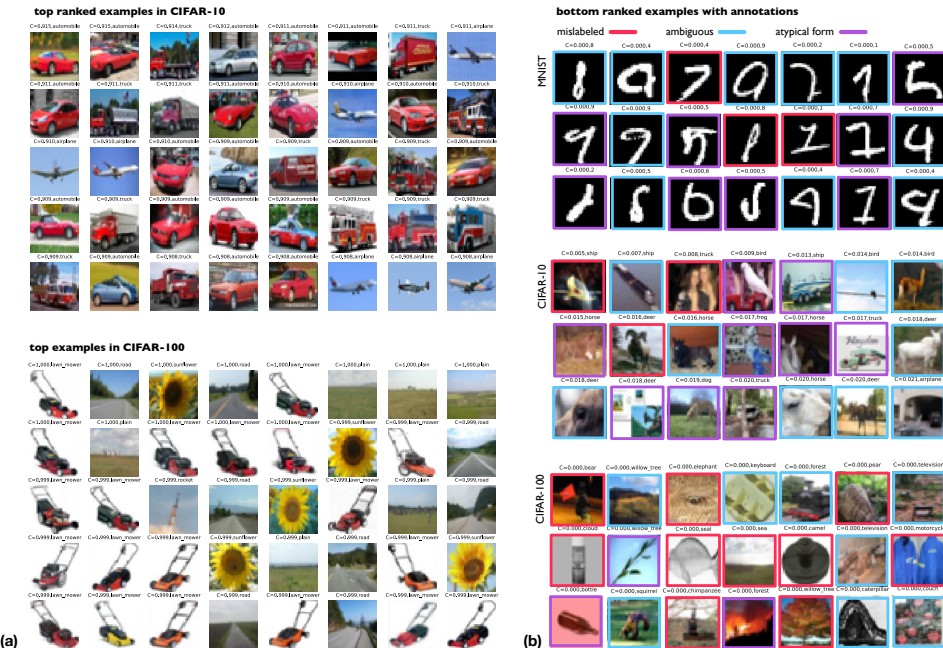

Figure 3: (a) Top ranked examples in CIFAR-10 and CIFAR-100. (b) Bottom ranked examples with annotations.

generally unknown for typical learning problems. In practice, we usually have a fixed data set $\hat{\mathcal{D}}$ consisting of $N$ i.i.d. samples from $\mathcal{P}$. So we can estimate the consistency profile with the following *empirical consistency profile*:

$$\hat{C}_{\hat{\mathcal{D}},n}(x,y) = \hat{\mathbb{E}}^r_{D \sim \hat{\mathcal{D}}^n} \left[ \mathbb{P}(f(x; D \backslash \{(x,y)\}) = y) \right], \quad n = 0, 1, \dots N-1 \tag{2}$$

where $D$ is a subset of size $n$ uniformly sampled from $\hat{\mathcal{D}}$ excluding $(x,y)$, and $\hat{\mathbb{E}}^r$ denotes empirical averaging with $r$ i.i.d. samples of such subsets. To obtain a reasonably accurate estimate (say, $r = 1000$), calculating the empirical consistency profile is still computationally prohibitive. For example, with each of the 50,000 training example in the CIFAR-10 training set, we need to train more than 2 trillion models. To obtain an estimate within the capability of current computation resources, we make two observations. First, model performance is generally stable when the training set size varies within a small range. Therefore, we can sample across the range of $n$ that we're concerned with and obtain the full profile via smooth interpolation. Second, let $D$ be a random subset of training data, then the single model $f(\cdot; D)$ can be reused in the estimation of all of the held-out examples $(x,y) \in \hat{\mathcal{D}} \backslash D$. As a result, with clever grouping and reuse, the number of models we need to train can be greatly reduced (See Algorithm 1 in the Appendix).

In particular, we sample $n$ dynamically according to the *subset ratio* $s \in \{10\%, \dots, 90\%\}$ of the full available training set. We sample 2,000 subsets for the empirical expectation of each $n$ and visualize the estimated consistency profiles for clusters of similar examples in Figure 2. One interesting observation is that while CIFAR-100 is generally more difficult than CIFAR-10, the top ranked examples (magenta lines) in CIFAR-100 are more likely to be classified correctly when the subset ratio is low. Figure 3a visualizes the top ranked examples from the two data sets. Note that in CIFAR-10, the dense modes from the truck and automobile classes are quite similar.

In contrast, Figure 2 indicates that the bottom-ranked examples (cyan lines) have persistently low probability of correct classification—sometimes below chance—even with a 90% subset ratio. We visualize some bottom-ranked examples and annotate them as (possibly) mislabeled, ambiguous (easily confused with another class or hard to identify the contents), and atypical form (e.g., burning "forest", fallen "bottle"). As the subset ratio grows, regularities in the data distribution systematically pull the ambiguous instances in the wrong direction. This behavior is analogous to the phenomenon we mentioned earlier that children over-regularize verbs (GO→GOED) as they gain more exposure to a language.

To get a total ordering of the examples in a data set, we distill the consistency profiles into a scalar *consistency score*, or C-score, by taking the expectation over $n$:

$$\hat{C}_{\hat{\mathcal{D}}}(x,y) = \mathbb{E}_n[\hat{C}_{\hat{\mathcal{D}},n}(x,y)] \tag{3}$$

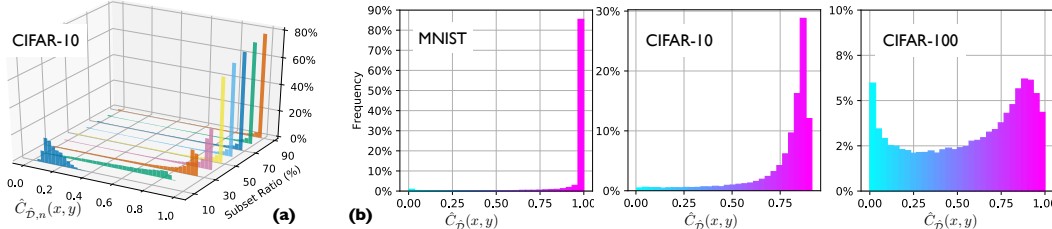

Figure 4: (a) Histogram of $\hat{C}_{\hat{\mathcal{D}},n}$ for each subset ratio on CIFAR-10. (b) Histogram of the C-score $\hat{C}_{\hat{\mathcal{D}}}$ averaged over all subset ratios on 3 different data sets.

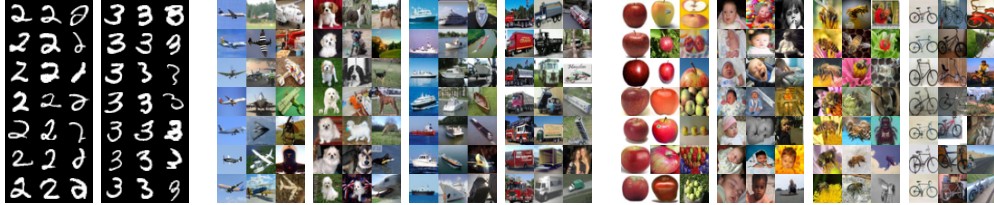

Figure 5: Examples from MNIST (blocks 1, 2), CIFAR-10 (blocks 3-6), and CIFAR-100 (blocks 7-10). Each block shows a single class; the left, middle, and right columns of a block depict instances with high, intermediate, and low C-scores, respectively.

For the case where $n$ is sampled according to the subset ratio $s$, the expectation is taken over a uniform distribution over sampled subset sizes.

## 4 THE STRUCTURAL REGULARITIES OF COMMON IMAGE DATA SETS

We apply the C-score estimate to analyze several common image data sets: MNIST, CIFAR-10, CIFAR-100, and ImageNet. See Appendix A for details on architectures and hyperparameters.

Figure 4a shows the distribution of $\hat{C}_{\hat{\mathcal{D}},n}$ on CIFAR-10 for the values of $n$ corresponding to each subset ratio $s \in \{10, ..., 90\}$. For each $s$, 2000 models are trained and held-out examples are evaluated. The Figure suggests that depending on $s$, instances may be concentrated near floor or ceiling, making them difficult to distinguish (as we elaborate further shortly). By taking an expectation over $s$, the C-score is less susceptible to floor and ceiling effects. Figure 4b shows the histogram of this integrated C-score on MNINT, CIFAR-10, and CIFAR-100. The histogram of CIFAR-10 in Figure 4b is distributed toward the high end, but is more uniformly spread than the histograms for specific subset ratios in Figure 4a.

For MNIST, CIFAR-10, and CIFAR-100, Figure 5 presents instances that vary in C-score. Each block of examples is one category; the left, middle, and right columns have high, intermediate, and low C-scores, respectively. The homogeneity of examples in the left column suggests dense modes generally consist of central cropped images of well aligned instances in their typical poses, with highly uniform color schemes. In contrast, many of the examples in the right column are in atypical forms or even ambiguous.

Next we apply the C-score analysis to the ImageNet data set. Training a standard model on ImageNet costs one to two orders of magnitude more computing resources than training on CIFAR, preventing us from running the C-score estimation procedure described early. Instead, we investigated the feasibility of approximating the C-score with a *point estimate*, i.e., selection of the $s$ that best represents the integral score. This is equivalent to taking expectation of $s$ with respect to a point-mass distribution, as opposed to the uniform distribution over subset ratios. By 'best represents,' we mean that the ranking of instances by the score matches the ranking by the score for a particular $s$.

Figure 6a shows the rank correlation between the integral score and the score for a given $s$, as a function of $s$ for our three smaller data sets, MNIST, CIFAR-10, and CIFAR-100. Examining the green CIFAR-10 curve, there is a peak at $s = 30$, indicating that $s = 30$ yields the best point-estimate approximation for the integral C-score. That the peak is at an intermediate $s$ is consistent with the observation from Figure 2 that the C-score bunches together instances for low and high $s$. For MNIST (blue curve), a less challenging data set than CIFAR-10, the peak is lower, at $s = 10$; for CIFAR-100 (orange curve), a more challenging data set than CIFAR-10, the peak is higher, at $s = 40$

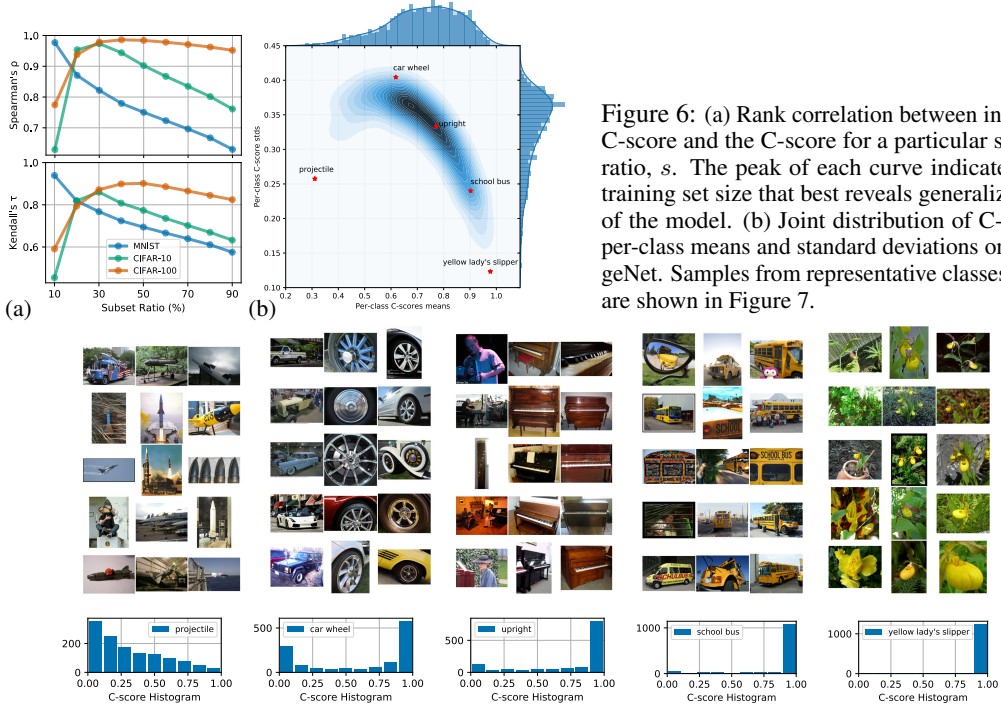

Figure 6: (a) Rank correlation between integral C-score and the C-score for a particular subset ratio, $s$. The peak of each curve indicates the training set size that best reveals generalization of the model. (b) Joint distribution of C-score per-class means and standard deviations on ImageNet. Samples from representative classes (★'s) are shown in Figure 7.

Figure 7: Example images from ImageNet. The 5 classes are chosen to have representative per-class C-score mean–standard-deviation profiles, as shown in Figure 6a. For each class, the three columns show sampled images from the (C-score ranked) top 99%, 35%, and 1% percentiles, respectively. The bottom pane shows the histograms of the C-scores in each of the 5 classes.

or $s = 50$. Thus, the peak appears to shift to larger $s$ for more challenging data sets. This finding is not surprising: more challenging data sets require a greater diversity of training instances in order to observe generalization.

Based on these observations, we picked $s = 70$ for a point estimate on ImageNet. In particular, we train 2,000 ResNet-50 models each with a random 70% subset of the ImageNet training set, and estimate the C-score based on those models.

The examples shown in Figure 1c are ranked according to this C-score estimate. Because ImageNet has 1,000 classes, we cannot offer a simple overview over the entire data set as in MNIST and CIFAR. Instead, we focus on analyzing the behaviors of individual classes. Specifically, we compute the mean and standard deviation (SD) of the C-scores of all the examples in a particular class. The mean C-scores indicates the relative difficulty of classes, and the SD indicates the diversity of examples within each class. The two-dimensional histogram in Figure 6a depicts the joint distribution of mean and SD across all classes. We selected several classes with various combinations of mean and SD, indicated by the ★'s in Figure 6a. We then selected sample images from the top 99%, 35% and 1% percentile ranked by the C-score within each class, and show them in Figure 7.

*Projectile* and *yellow lady's slipper* represent two extreme cases of diverse and unified classes, respectively. Most other classes lie in the high density region of the 2D histogram in Figure 6b, and share a common pattern of a densely populated mode of highly regular examples and a tail of rare, ambiguous examples. The tail becomes smaller from the class *car wheel* to *upright* and *school bus*.

## 5 ANALYZING LEARNING DYNAMICS WITH THE C-SCORE

In this section, we use the C-score to study the behavior of different optimizers for training deep neural networks. For this study, we partition the CIFAR-10 training set into subsets by C-score. Then we record the learning curves—model accuracy over training epochs—for each set. Figure 8 plots the learning curves for C-score-binned examples. The left panel shows SGD training with a stagewise constant learning rate, and the right panel shows the Adam optimizer (Kingma & Ba, 2015), which scales the learning rate adaptively. In both cases, the groups with high C-scores (magenta) generally learn faster than the groups with low C-scores (cyan). Intuitively, the high C-score groups consist of mutually consistent examples that support one another during training, whereas the low C-score

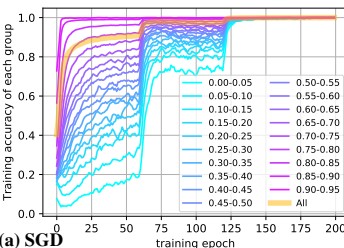 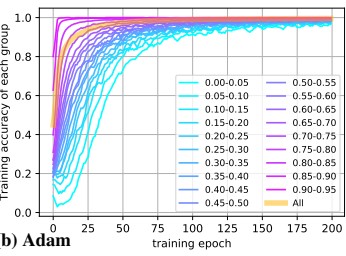

**(a) SGD**     **(b) Adam**

Figure 8: Learning speed of CIFAR-10 examples grouped by C-score. The thick transparent curve shows the average accuracy over the entire training set. SGD achieves test accuracy 95.14%, Adam achieves 92.97%.

groups consist of irregular examples forming sparse modes with fewer consistent peers. In the case of true outliers, the model needs to memorize the labels individually as they do not share structure with any other examples.

The learning curves have wider dispersion in SGD than in Adam. Early in SGD training where the learning rate is large, the examples with the lowest C-scores barely learn. Figure 9a shows SGD with constant learning rate corresponding to values used in each of the learning rate stages in Figure 8a, confirming that with a large learning rate, the groups with the lowest C-scores are not learned well even after the full 200 epochs of training. In comparison, Adam shows less spread among the groups and as a result, converges sooner. However, the superior convergence speed of adaptive optimizers like Adam does not always lead to better generalization (Wilson et al., 2017; Keskar & Socher, 2017; Luo et al., 2019). We observe this outcome as well: SGD with a stagewise learning rate achieves 95.14% test accuracy, compared to 92.97% for Adam.

To develop our intuitions for this phenomenon, we are assisted by the C-score partitioning of examples in Figure 8. Starting training with a large learning rate effectively enforces a sort of curriculum in which the model first learns only the strongest regularities. At a later stage when the learning rate is lowered and exceptions or outliers are able to be learned, the model has already built a solid representation based on domain regularities. However, when training starts with a small learning rate or uses Adam to adaptively scale the learning rate, all the examples are learned at similar pace. In this case, it is difficult to build simple and coherent representations that need to account for all different modes even in the early stage of learning, which eventually lead to a model with less generalization power.

# 6 C-SCORE PROXIES

We are able to reduce the cost of estimating C-scores from infeasible to feasible, but the procedure is still very expensive. Ideally, we would like to have more efficient *proxies* that do not require training multiple models. We use the term *proxy* to refer to any quantity that is well correlated with the C-score but does not have a direct mathematical relation to it, as contrasted with *approximations* that are designed to mathematically approximate the C-score (e.g., approximating the expectation with empirical averaging). The possible candidate set for C-score proxies is very large, as any measure that reflects information about difficulty or regularity of examples could be considered. Our Related Work section mentions a few such possibilities. We examined a collection of proxies based on inter-example distances in input and latent space, but none were terribly promising. (See Appendix for details.)

Inspired by our observations in the previous section that the speed-of-learning tends to correlate with the C-score rankings, we instead focus on a class of learning-speed based proxies that have the added bonus of being trivial to compute. Intuitively, a training example that is consistent with many others should be learned quickly because the gradient steps for all consistent examples should be well aligned. One might therefore conjecture that strong regularities in a data set are not only better learned at asymptote—leading to better generalization performance—but are also learned *sooner* in the time course of training. This *learning speed* hypothesis is nontrivial, because the C-score is defined for a held-out instance following training, whereas learning speed is defined for a training instance during training. This hypothesis is qualitatively verified from Figure 8. In particular, the cyan examples having the lowest C-scores are learned most slowly and the purple examples having the highest C-scores are learned most quickly. Indeed, learning speed is monotonically related to C-score bin.

Figure 9b shows a quantitative evaluation, where we compute the Spearman's rank correlation between the C-score of an instance and various proxy scores based on learning speed. In particular, we test *accuracy* (0-1 correctness), $p_L$ (softmax confidence on the correct class), $p_{\max}$ (max softmax confidence across all classes) and *entropy* (negative entropy of softmax confidences). We use

Figure 9: (a) Learning speed of CIFAR-10 examples grouped by C-score with SGD using constant learning rate. The 4 learning rates correspond to the constants in the stage-wise scheduler in Figure 8a. The test accuracies for those models are 84.84%, 91.19%, 92.05%, and 90.82%, respectively. (b) Rank correlation (Spearman's $\rho$) between C-score and training statistics based proxies. (c) Using C-score proxies to identify outliers on CIFAR-10.

*cumulative* statistics which average from the beginning of training to the current epoch because the cumulative statistics yield a more stable measure—and higher correlation—than statistics based on a single epoch. We also compare to a *forgetting-event* statistic (Toneva et al., 2019), which is simply a count of the number of transitions from "learned" to "forgotten" during training. All of these proxies show strong correlation with the C-score. $p_L$ reaches $\rho \approx 0.9$ at the peak. $p_{max}$ and *entropy* perform similarly, both slightly worse than $p_L$. The plot also shows that examples with low cscores are more likely to be forgotten during training. The *forgetting event* based proxy slightly under-performs other proxies and takes a larger number of training epochs to reach its peak correlation. We suspect this is because forgetting events happen only *after* an example is learned, so unlike other proxies studied here, forgetting statistics for hard examples cannot be obtained in the earlier stage of training.

Finally, we present a simple demonstration of the utility of our proxies for outlier detection. We corrupt a fraction $\gamma = 25\%$ of the CIFAR-10 training set with random label assignments. Then during training, we identify the fraction $\gamma$ with the lowest ranking by three C-score proxies—cumulative accuracy, $p_L$, and forgetting-event statistics. Figure 9c shows the removal rate—the fraction of the lowest ranked examples which are indeed outliers; two of the C-score proxies successfully identify over 95% of outliers.

# 7 DISCUSSION

We formulated a *consistency profile* for individual examples in a data set that reflects the probability of correct generalization to the example as a function of training set size. This profile has strong ties to generalization theory and matches basic intuitions about data regularity in both human and machine learning. We distilled the profile into a scalar C-score, which provides a total ordering of the instances in a data set by essentially the sample complexity—the amount of training data required—to ensure correct generalization to the instance. To leverage the C-score to analyze structural regularities in complex data sets, we derived a C-score estimation procedure and obtained C-scores for examples in MNIST, CIFAR-10, CIFAR-100, and ImageNet. The C-score estimate helps to characterize the continuum between a densely populated mode consisting of aligned, centrally cropped examples with unified shape and color profiles, and sparsely populated modes of just one or two instances. We also used the C-score as a tool to compare the learning dynamics of different optimizers to better understand the consequences for generalization. To make the C-score useful in practice for new data sets, we explored a number of efficient proxies for the C-score based on the learning speed, and found that the integrated accuracy over training is a strong indicator of an example's ranking by C-score, which reflects generalization accuracy had the example been held out from the training set.

In the 1980s, neural nets were touted for learning *rule-governed behavior* without explicit rules (Rumelhart & McClelland, 1986). At the time, AI researchers were focused on constructing expert systems by extracting explicit rules from human domain experts. Expert systems ultimately failed because the diversity and nuance of statistical regularities in a domain was too great for any human to explicate. In the modern deep learning era, researchers have made much progress in automatically extracting regularities from data. Nonetheless, there is still much work to be done to understand these regularities, and how the consistency relationships among instances determine the outcome of learning. By defining and investigating a consistency score, we hope to have made some progress in this direction.

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

## A EXPERIMENT DETAILS

The details on model architectures, data set information and hyper-parameters used in the experiments for empirical estimation of the C-score can be found in Table 1. We implement our experiment in Tensorflow (Abadi et al., 2015). The holdout subroutine used in C-score estimation is listed in Algorithm 1. Most of the training jobs for C-score estimation are run on single NVidia® Tesla P100 GPUs. The ImageNet training jobs are run with 8 P100 GPUs using single-node multi-GPU data parallelization.

The experiments on learning speed are conducted with ResNet-18 on CIFAR-10, trained for 200 epochs while batch size is 32. For optimizer, we use the SGD with the initial learning rate 0.1, momentum 0.9 (with Nesterov momentum) and weight decay is 5e-4. The stage-wise constant learning rate scheduler decrease the learning rate at the 60th, 90th, and 120th epoch with a decay factor of 0.2.

---

**Algorithm 1** Estimation of $\hat{C}_{\hat{\mathcal{D}},n}$

---

**Input:** Data set $\hat{\mathcal{D}} = (X, Y)$ with $N$ examples
**Input:** $n$: number of instances used for training
**Input:** $k$: number of subset samples
**Output:** $\hat{C} \in \mathbb{R}^N$: $(\hat{C}_{\hat{\mathcal{D}},n}(x,y))_{(x,y)\in\hat{\mathcal{D}}}$
   Initialize binary mask matrix $M \leftarrow 0^{k \times N}$
   Initialize 0-1 loss matrix $L \leftarrow 0^{k \times N}$
   **for** $i \in (1, 2, \ldots, k)$ **do**
      Sample $n$ random indices $I$ from $\{1, \ldots, N\}$
      $M[i, I] \leftarrow 1$
      Train $\hat{f}$ from scratch with the subset $X[I], Y[I]$
      $L[i, :] \leftarrow \mathbf{1}[\hat{f}(X) \neq Y]$
   **end for**
   Initialize score estimation vector $\hat{C} \leftarrow 0^N$
   **for** $j \in (1, 2, \ldots, N)$ **do**
      $Q \leftarrow \neg M[:, j]$
      $\hat{C}[j] \leftarrow \text{sum}(\neg L[:, Q])/\text{sum}(Q)$
   **end for**

---

## B TIME AND SPACE COMPLEXITY

The time complexity of the holdout procedure for empirical estimation of the C-score is $\mathcal{O}(S(kT + E))$. Here $S$ is the number of subset ratios, $k$ is number of holdout for each subset ratio, and $T$ is the average training time for a neural network. $E$ is the time for computing the score given the $k$-fold holdout training results, which involves elementwise computation on a matrix of size $k \times N$, and is negligible comparing to the time for training neural networks. The space complexity is the space for training a single neural network times the number of parallel training jobs. The space complexity for computing the scores is $\mathcal{O}(kN)$.

For kernel density estimation based scores, the most expensive part is forming the pairwise distance matrix (and the kernel matrix), which requires $\mathcal{O}(N^2)$ space and $\mathcal{O}(N^2d)$ time, where $d$ is the dimension of the input or hidden representation spaces.

## C MORE VISUALIZATIONS OF IMAGES RANKED BY C-SCORE

Examples with high, middle and low C-scores from a few representative classes of MNIST, CIFAR-10 and CIFAR-100 are show in Figure 5. In this appendix, we depict the results for all the 10 classes of MNIST and CIFAR-10 in Figure 10 and Figure 11, respectively. The results from the first 60 out of the 100 classes on CIFAR-100 is depicted in Figure 12. Figure 13 and Figure 14 show more examples from ImageNet. Please see (URL anonymized) for more visualizations.

| | MNIST | CIFAR-10 | CIFAR-100 | ImageNet |
|---|---|---|---|---|
| Architecture | MLP(512,256,10) | Inception[†] | Inception[†] | ResNet-50 (V2) |
| Optimizer | SGD | SGD | SGD | SGD |
| Momentum | 0.9 | 0.9 | 0.9 | 0.9 |
| Base Learning Rate | 0.1 | 0.4 | 0.4 | $0.1 \times 7$ |
| Learning Rate Scheduler | $\wedge(15\%)^\star$ | $\wedge(15\%)^\star$ | $\wedge(15\%)^\star$ | LinearRampupPiecewiseConstant[⋆⋆] |
| Batch Size | 256 | 512 | 512 | $128 \times 7$ |
| Epochs | 20 | 80 | 160 | 100 |
| Data Augmentation | · · · · · · Random Padded Cropping⊛ + Random Left-Right Flipping · · · · · · | | | |
| Image Size | $28 \times 28$ | $32 \times 32$ | $32 \times 32$ | $224 \times 224$ |
| Training Set Size | 60,000 | 50,000 | 50,000 | 1,281,167 |
| Number of Classes | 10 | 10 | 100 | 1000 |

†    A simplified Inception model suitable for small image sizes, defined as follows:
       Inception :: Conv($3 \times 3$, 96) → Stage1 → Stage2 → Stage3 → GlobalMaxPool → Linear.
         Stage1 :: Block(32, 32) → Block(32, 48) → Conv($3 \times 3$, 160, Stride=2).
         Stage2 :: Block(112, 48) → Block(96, 64) → Block(80, 80) → Block (48, 96) → Conv($3 \times 3$,
240, Stride=2).
         Stage3 :: Block(176, 160) → Block(176, 160).
     Block($C_1$, $C_2$) :: Concat(Conv($1 \times 1$, $C_1$), Conv($3 \times 3$, $C_2$)).
         Conv :: Convolution → BatchNormalization → ReLU.

⋆    $\wedge(15\%)$ learning rate scheduler linearly increase the learning rate from 0 to the *base learning rate* in the first 15% training steps, and then from there linear decrease to 0 in the remaining training steps.

⋆⋆    LinearRampupPiecewiseConstant learning rate scheduler linearly increase the learning rate from 0 to the *base learning rate* in the first 15% training steps. Then the learning rate remains piecewise constant with a $10\times$ decay at 30%, 60% and 90% of the training steps, respectively.

⊛    Random Padded Cropping pad 4 pixels of zeros to all the four sides of MNIST, CIFAR-10, CIFAR-100 images and (randomly) crop back to the original image size. For ImageNet, a padding of 32 pixels is used for all four sides of the images.

Table 1: Details for the experiments used in the empirical estimation of the C-score.

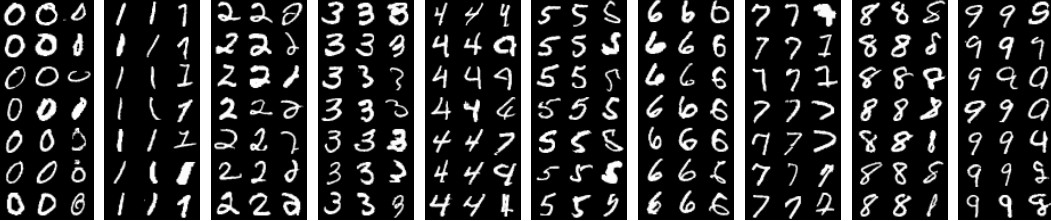

Figure 10: Examples from MNIST. Each block shows a single class; the left, middle, and right columns of a block depict instances with high, intermediate, and low C-scores, respectively.

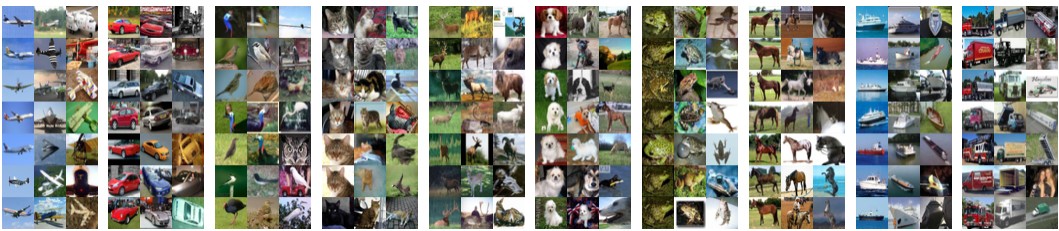

Figure 11: Examples from CIFAR-10. Each block shows a single class; the left, middle, and right columns of a block depict instances with high, intermediate, and low C-scores, respectively.

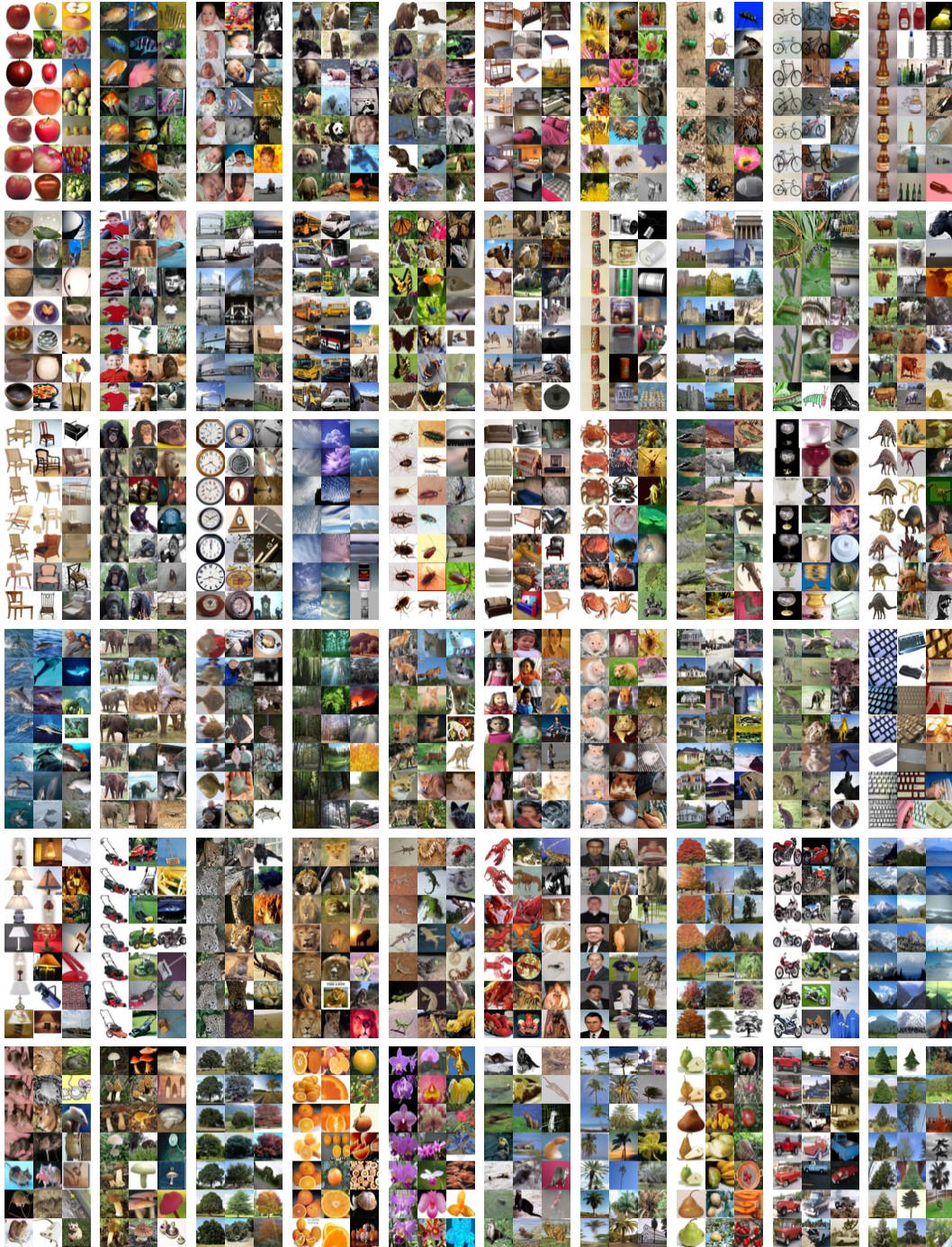

Figure 12: Examples from CIFAR-100. Each block shows a single class; the left, middle, and right columns of a block depict instances with high, intermediate, and low C-scores, respectively. The first 60 (out of the 100) classes are shown.

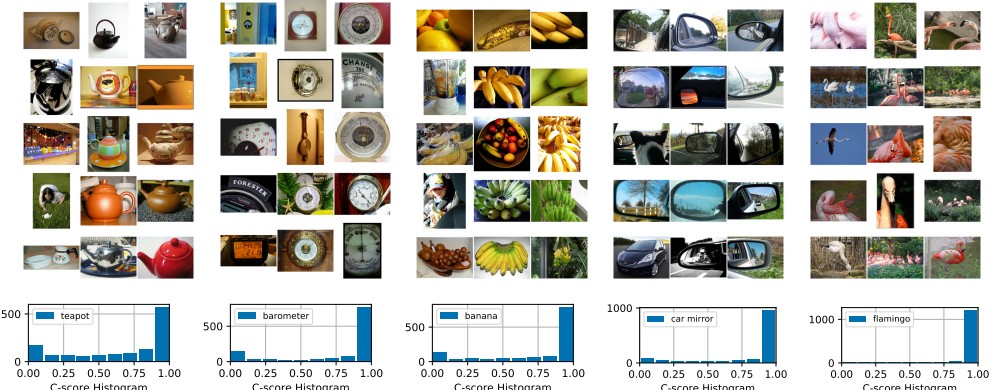

Figure 13: Example images from ImageNet. For each class, the three columns show sampled images from the (C-score ranked) top 99%, 35%, and 1% percentiles, respectively. The bottom pane shows the histograms of the C-scores in each of the 5 classes.

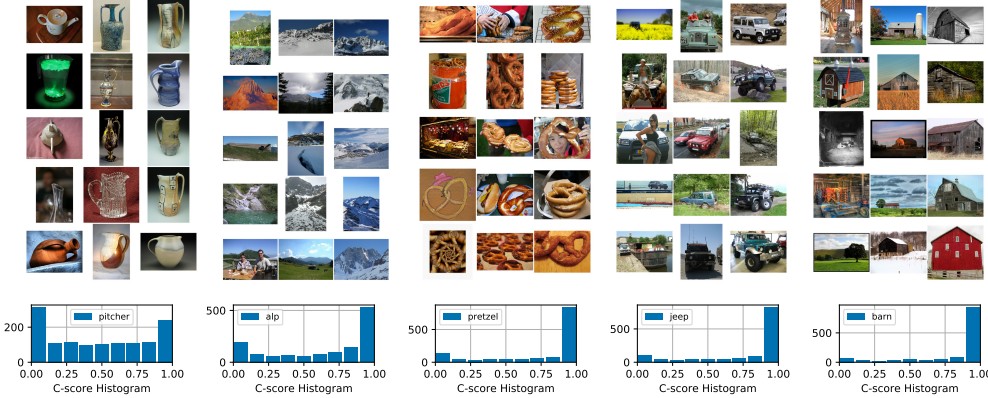

Figure 14: Example images from ImageNet. For each class, the three columns show sampled images from the (C-score ranked) top 99%, 35%, and 1% percentiles, respectively. The bottom pane shows the histograms of the C-scores in each of the 5 classes.

|   |           | $\hat{C}$ | $\hat{C}^L$ | $\hat{C}^{\pm L}$ | $\hat{C}^{\text{LOF}}$ |
|---|-----------|-----------|-------------|-------------------|------------------------|
| $\rho$ | CIFAR-10  | $-0.064$  | $-0.009$    | $0.083$           | $0.103$                |
|        | CIFAR-100 | $-0.098$  | $0.117$     | $0.105$           | $0.151$                |
| $\tau$ | CIFAR-10  | $-0.042$  | $-0.006$    | $0.055$           | $0.070$                |
|        | CIFAR-100 | $-0.066$  | $0.078$     | $0.070$           | $0.101$                |

Table 2: Rank correlation between C-score and pairwise distance based proxies on inputs. Measured with Spearman's $\rho$ and Kendall's $\tau$ rank correlations, respectively.

## D  C-SCORE PROXIES BASED ON PAIRWISE DISTANCES

We study C-score proxies based on pairwise distances here. Intuitively, an example is consistent with the data distribution if it lies near other examples having the same label. However, if the example lies far from instances in the same class or lies near instances of different classes, one might not expect it to generalize. Based on this intuition, we define a relative local-density score:

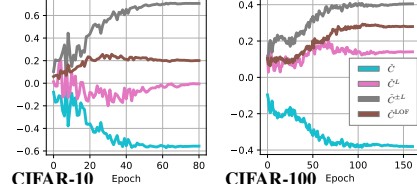

$$\hat{C}^{\pm L}(x,y) = \frac{1}{N}\sum_{i=1}^{N}2(\mathbf{1}[y=y_i]-\tfrac{1}{2})K(x_i,x), \quad (4)$$

Figure 15: Spearman rank correlation between C-score and distance-based score on hidden representations.

where $K(x,x') = \exp(-\|x-x'\|^2/h^2)$ is an RBF kernel with the bandwidth $h$, and $\mathbf{1}[\cdot]$ is the indicator function. To evaluate the importance of explicit label information, we study two related scores: $\hat{C}^L$ that uses only same-class examples when estimating the local density, and $\hat{C}$ that uses all the neighbor examples by ignoring the labels.

$$\hat{C}^L(x,y) = \frac{1}{N}\sum_{i=1}^{N}\mathbf{1}[y=y_i]K(x_i,x), \quad (5)$$

$$\hat{C}(x) = \frac{1}{N}\sum_{i=1}^{N}K(x_i,x). \quad (6)$$

We also study a proxy based on the local outlier factor (LOF) algorithm (Breunig et al., 2000), which measures the local deviation of each point with respect to its neighbours. Since large LOF scores indicate outliers, we use the negative LOF score as a C-score proxy, denoted by $\hat{C}^{\text{LOF}}(x)$.

Table 2 shows the agreement between the proxy scores and the estimated C-score. Agreement is quantified by two rank correlation measures on three data sets. As anticipated, the input-density score that ignores labels, $\hat{C}(x)$, and the class-conditional density, $\hat{C}^L(x,y)$, have poor agreement. $\hat{C}^{\pm L}(x,y)$ and $\hat{C}^{\text{LOF}}$ are slightly better. However, none of the proxies has high enough correlation to be useful, because it is very hard to obtain semantically meaningful distance estimations from the raw pixels.

Since proxies based on pairwise distances in the input space work poorly, we further evaluate the proxies using the penultimate layer of the network as a representation of an image: $\hat{C}_h^{\pm L}$, $\hat{C}_h^L$, $\hat{C}_h$ and $\hat{C}_h^{\text{LOF}}$, with the subscript $h$ indicating that the score operates in hidden space. For each score and data set, we compute Spearman's rank correlation between the proxy score and the C-score.

In particular, we train neural network models with the same specification as in Table 1 on the full training set. We use an RBF kernel $K(x,x') = \exp(-\|x-x'\|^2/h^2)$, where the bandwidth parameter $h$ is adaptively chosen as $1/2$ of the mean pairwise Euclidean distance across the data set. For the local outlier factor (LOF) algorithm (Breunig et al., 2000), we use the neighborhood size $k = 3$. See Figure 16 for the behavior of LOF across a wide range of neighborhood sizes.

Because the embedding changes as the network is trained, we plot the correlation as a function of training epoch in Figure 15. For both data sets, the proxy score that correlates best with the C-score is $\hat{C}_h^{\pm L}$ (grey), followed by $\hat{C}_h^{\text{LOF}}$ (brown), then $\hat{C}_h^L$ (pink) and $\hat{C}_h$ (blue). Clearly, appropriate use of

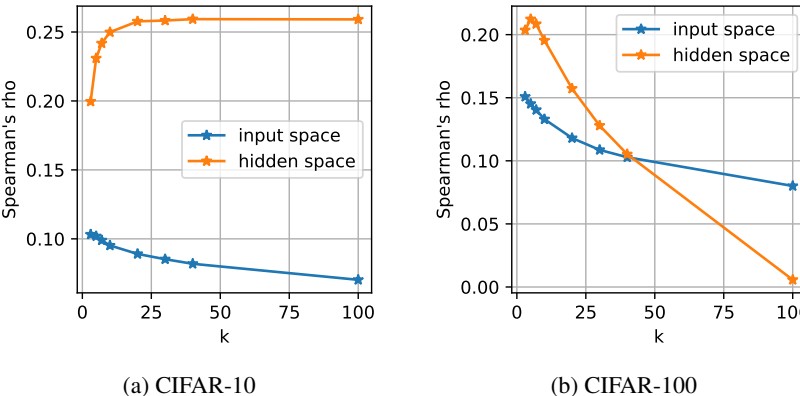

(a) CIFAR-10           (b) CIFAR-100

Figure 16: The Spearman's $\rho$ correlation between the C-score and the score based on LOF with different neighborhood sizes.

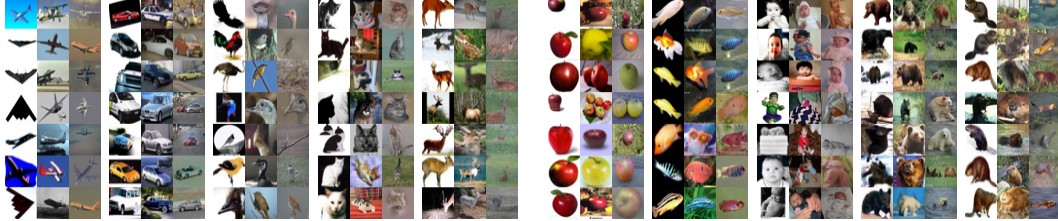

Figure 17: Examples from CIFAR-10 (left 5 blocks) and CIFAR-100 (right 5 blocks). Each block shows a single class; the left, middle, and right columns of a block depict instances with top, intermediate, and bottom ranking according to the relative local density score $\hat{C}^{\pm L}$ in the input space, respectively.

labels helps with the ranking. However, our proxy $\hat{C}_h^{\pm L}$ uses the labels in an ad hoc manner. We will discuss a more principled measure based on gradient vectors shortly and relate it to the neural tangent kernel Jacot et al. (2018).

The results reveal interesting properties of the hidden representation. One might be concerned that as training progresses, the representations will optimize toward the classification loss and may discard inter-class relationships that could be potentially useful for other downstream tasks (Scott et al., 2018). However, our results suggest that $\hat{C}_h^{\pm L}$ does not diminish as a predictor of the C-score, even long after training converges. Thus, at least some information concerning the relation between different examples is retained in the representation, even though intra- and inter-class similarity is not very relevant for a classification model. To the extent that the hidden representation—crafted through a discriminative loss—preserves class structure, one might expect that the C-score could be predicted without label reweighting; however, the poor performance of $\hat{C}_h$ suggests otherwise.

Figure 17 and Figure 18 visualize examples in CIFAR-10/CIFAR-100 ranked by the class weighted local density scores in the input and learned hidden space, respectively. The ranking calculated in the input space relies heavily on low level features that can be derived directly from the pixels like strong silhouette. The rankings calculated from the learned hidden space correlate better with C-score, though the visualization shows that the ranking are sometimes still noisy even for the top ranking examples (e.g. the class "automobile" in CIFAR-10).

### D.1 Pairwise Distance Estimation with Gradient Representations

Most modern neural networks are trained with first order gradient descent based algorithms and variants. In each iteration, the gradient of loss on a mini-batch of training examples evaluated at the current network weights is computed and used to update the current parameter. Let $\nabla_t(\cdot)$ be the function that maps an input-label training pair (the case of mini-batch size one) to the corresponding

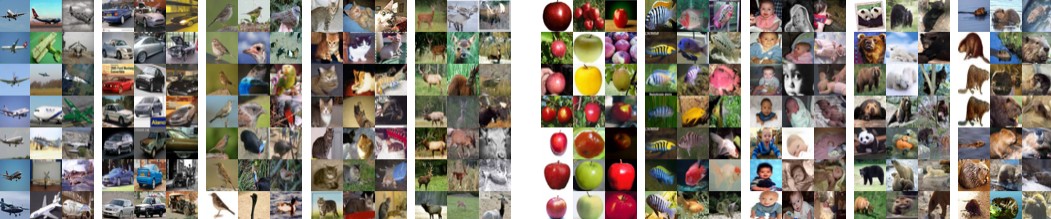

Figure 18: Examples from CIFAR-10 (left 5 blocks) and CIFAR-100 (right 5 blocks). Each block shows a single class; the left, middle, and right columns of a block depict instances with top, intermediate, and bottom ranking according to the relative local density score $\hat{C}_h^{\pm L}$ in the latent representation space of a trained network, respectively.

gradient evaluated at the network weights of the $t$-th iteration. Then this defines a gradient based representation on which we can compute density based ranking scores. The intuition is that in a gradient based learning algorithm, an example is consistent with others if they all compute similar gradients.

Comparing to the hidden representations defined the outputs of a neural network layer, the gradient based representations induce a more natural way of incorporating the label information. In the previous section, we reweight the neighbor examples belonging to a different class by 0 or -1. For gradient based representations, no ad hoc reweighting is needed as the gradient is computed on the loss that has already takes the label into account. Similar inputs with different labels automatically lead to dissimilar gradients. Moreover, this could seamlessly handle labels and losses with rich structures (e.g. image segmentation, machine translation) where an effective reweighting scheme is hard to find. The gradient based representation is closely related to recent developments on Neural Tagent Kernels (NTK) (Jacot et al., 2018). It is shown that when the network width goes to infinity, the neural network training dynamics can be effectively approximately via Taylor expansion at the initial network weights. In other words, the algorithm is effectively learning a *linear* model on the *nonlinear* representations defined by $\nabla_0(\cdot)$. This feature map induces the NTK, and connects deep learning to the literature of kernel machines.

Although NTK enjoys nice theoretical properties, it is challenging to perform density estimation on it. Even for the more practical case of *finite width* neural networks, the gradient representations are of extremely high dimensions as modern neural networks general have parameters ranging from millions to billions (e.g. Tan & Le, 2019; Radford et al., 2019). As a result, both computation and memory requirements are prohibitive if a naive density estimation is to be computed on the gradient representations. We leave as future work to explore efficient algorithms to practically compute this score.

## E    WHAT MAKES AN ITEM REGULAR OR IRREGULAR?

The notion of regularity is primarily coming from the statistical consistency of the example with the rest of the population, but less from the intrinsic structure of the example's contents. To illustrate this, we refer back to the experiments in Section 5 on measuring the learning speed of groups of examples generated via equal partition on the C-score value range $[0, 1]$. As shown in Figure 4b, the distribution is uneven between high and low C-score values. As a result, the high C-score groups will have more examples than the low C-score groups. This agrees with the intuition that regularity arises from high probability masses.

To test whether an example with top-ranking C-score is still highly regular after the density of its neighborhood is reduced, we redo the experiment, but subsample each group to contain an equal number ($\sim 400$) of examples. Then we run training on this new data set and observe the learning speed in each (subsampled) group. The result is shown in Figure 19, which is to be compared with the results without group-size-equalizing in Figure 8a in the main text. The following observations can be made:

1.  The learning curves for many of the groups start to overlap with each other.

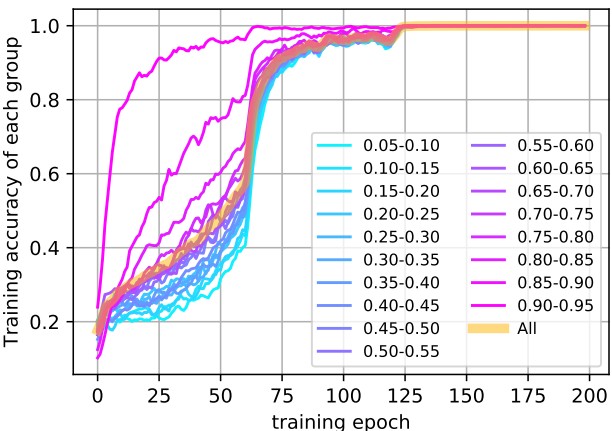

Figure 19: Learning speed of group of examples ranked by C-scores, with equal number (400) of examples in each group via subsampling.

2. The lower ranked groups now learns faster. For example, the lowest ranked group goes above 30% accuracy near epoch 50. In the original experiment (Figure 8a), this groups is still below 20% accuracy at epoch 50. The model is now learning with a much smaller data set. Since the lower ranked examples are not highly consistent with the rest of the population, this means there are fewer "other examples" to compete with (i.e. those "other examples" will move the weights towards a direction that is less preferable for the lower ranked examples). As a result, the lower ranked groups can now learn faster.

3. On the other hand, the higher ranked groups now learn slower, which is clear from a direct comparison between Figure 8a and Figure 19. This is because for highly regular examples, reducing the data set size means removing consistent examples — that is, there are now less "supporters" as oppose to less "competitors" in the case of lower ranked groups. As a result, the learn speed is now slower.

4. Even though the learning curves are now overlapping, the highest ranked group and the lowest ranked group are still clearly separated. The potential reason is that while the lower ranked examples can be outliers in many different ways, the highest ranked examples are probably regular in a single (or very few) visual clusters (see the top ranked examples in Figure 5). As a result, the within group diversities of the highest ranked groups are still much smaller than the lowest ranked groups.

In summary, the regularity of an example arises from its consistency relation with the rest of the population. A regular example in isolation is no different to an outlier. Moreover, it is also not merely an intrinsic property of the data distribution, but is closely related to the model, loss function and learning algorithms. For example, while a picture with a red lake and a purple forest is likely be considered an outlier in the usual sense, for a model that only uses grayscale information it could be highly regular.

## F  SENSITIVITY OF C-SCORES TO THE NUMBER OF MODELS

We used 2,000 models per subset ratio to evaluate C-scores in our experiments to ensure that we get stable estimates. In this section, we study the sensitivity of C-scores with respect to the number of models and evaluate the possibility to use fewer models in practice. Let $C_{0-2k}$ be the C-scores estimated with the full 2,000 models per subset ratio. We split the 2,000 models for each subset ratio into two halves, and obtain two independent estimates $C_{0-1k}$ and $C_{1k-2k}$. Then for $m \in \{1, 2, 4, 8, 16, 32, 64, 128, 256, 512, 1000\}$, we sample $m$ random models from the first 1,000 split, and estimate C-scores (denoted by $C_m$) based on those models. We compute the Spearman's $\rho$ correlation between each $C_m$ and $C_{1k-2k}$. The results are plotted in Figure 20. The random sampling of $m$ models is repeated 10 times for each $m$ and the error bars show the standard deviations. The

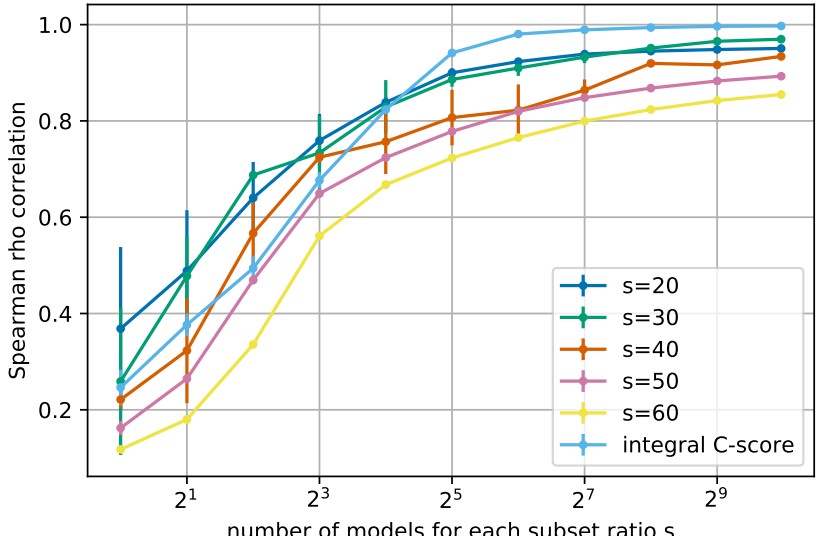

Figure 20: The correlation of C-scores estimated with varying numbers of models (the x-axis) and C-scores estimated with 1,000 independent models. The simulations are run with CIFAR-10, and the error bars show standard deviation from 10 runs.

figure shows that a good correlation is found for as few as $m = 64$ models. However, the integral C-score requires training models for various subset ratios (9 different subset ratios in our simulations), so the total number of models needed is roughly $64 \times 9$. If we want to obtain a reliable estimate of the C-score under a single fixed subset ratio, we find that we need 512 models in order to get a $> .95$ correlation with $C_{1k-2k}$. So it appears that whether we are computing the integral C-score or the C-score for a particular subset ratio, we need to train on the order of 500-600 models.

In the analysis above, we have used $C_{1k-2k}$ as the reference scores to compute correlation to ensure no overlapping between the models used to compute different estimates. Note $C_{1k-2k}$ itself is well correlated with the the full estimate from 2,000 models, as demonstrated by the following correlations: $\rho(C_{0-1k}, C_{1k-2k}) = 0.9996$, $\rho(C_{0-1k}, C_{0-2k}) = 0.9999$, and $\rho(C_{1k-2k}, C_{0-2k}) = 0.9999$.

## G  CODE AND PRE-COMPUTED C-SCORES

We provide code implementing our C-score estimation algorithms, and pre-computed C-scores and associated model checkpoints for CIFAR-10, CIFAR-100 and ImageNet at (URL anonymized). The exported files are in Numpy's data format saved via `numpy.savez`. For CIFAR-10 and CIFAR-100, the exported file contains two arrays `labels` and `scores`. Both arrays are stored in the order of training examples as defined by the original data sets found at `https://www.cs.toronto.edu/~kriz/cifar.html`. The data loading tools provided in some deep learning library might not be following the original data example orders, so we provided the `labels` array for easy sanity check of the data ordering.

For ImageNet, since there is no well defined example ordering, we order the exported scores arbitrarily, and include a script to reconstruct the data set with index information by using the filename of each example to help identify the example-score mapping.

