# OpenReview forum: "Characterizing Structural Regularities of Labeled Data in Overparameterized Models"
_ICLR.cc/2021/Conference — Reject_

### Official Review · AnonReviewer2 · 2020-10-25
**Intuitive measure but not enough practical implications**

**Rating:** 5
**Confidence:** 4

**Review:**

Summary:
This paper formulates a consistency score (C-score, which characterizes
the expected accuracy for a held-out instance given training sets of varying size
sampled from the data distribution) to measure the regularity of an example. It also proposes approximations for C-score and study structural regularities of MNIST, CIFAR-10, CIFAR-100, and ImageNet. C-score is also used to learn dynamics of different optimizers and consequences for generalization. It further provides proxies (in particular, the learning speed) for C-score for practical tasks such as outlier-detection.

################################################

Reasons for score:
The paper is overall well-written and presents some interesting findings about a newly proposed measure C-score. However, the novelty of the consistency score and the experiments on utilizing it for outlier detection are relatively limited.


################################################

Pros:

+well-written and easy to follow

+interesting observations and good visualizations

+experiments in general support claims

Cons:

-Not enough results shown to demonstrate how useful this measure is and no baselines are provided.

-The idea of regular/irregular is not new.

-Pure empirical findings; more discussions on the theoretical side is needed


One of my main concerns is the practical implication of the proposed measure. Only a simple mislabelling experiment on one model and dataset is shown and no baseline methods are compared with.

The idea of attributing examples into regular examples and irregular examples is also not new. The learning speed and generalization as the authors mentioned has been previously observed. Using C-score to analyze the dynamics of different optimizers is interesting but is not explored deeper / more extensively.

################################################

Questions:

-Why choosing s=70 for ImageNet? rather than 50 or other numbers? Since ImageNet is much more complex than CIFAR, it is hard to tell if similar observations (other than the examples given) hold.

-Ablation study on the number of models (2000 is used currently for all experiments) to train for each subset ratio should be provided at least for one dataset.

-I believe the code can be made public anonymously?

################################################

Post-Rebuttal:

I want to thanks the authors for their detailed responses. It addresses my concern of hyperparameter choices. However, after going through the responses and paper again, I decide to maintain my initial assessment. The main reasons are that 1.conceptually C-score is not very novel. The addition of this paper to our existing understanding is not that much. 2.practically the C-score does not improve the-state-of-art of outlier detection (comparison with traditional methods is lacking).

I encourage the authors to further explore using C-score on improving outlier detection and comparing with existing traditional methods. Besides, diving a bit deeper into using C-score to analyze the dynamics of different optimizers will be very interesting.

---

> ### Author Response · Authors · 2020-11-19
> **Thanks for the comments!**
>
> **Idea of regular/irregular is not new**
> We appreciate that the notion of regularity/irregularity of examples is not new. Indeed, it has been the topic of early first studies of neural net learning (e.g., Rumelhart & McClelland, 1986, cited in our paper). Our contribution is the formulation of an analytical tool for a quantitative investigation of regularity/irregularity.  One indication of this value is its impact on research. Since publicizing our work, we have had requests from 9 distinct groups of researchers wishing to access the C-scores for particular data sets in order to further their own research agendas. For example, one group is studying the temporal dynamics of recognition in recurrent/feedback perceptual models and the relation between the time to settle on a response and the C-score (finding a significant correlation).
>
> **The learning speed and generalization as the authors mentioned has been previously observed**
> Although we made this statement in our related work section, we now realize that we overstated past results, and have corrected the paper. We cite 3 related articles, but none of them make the same point that we do -- that individual examples which are learned more quickly during training are likely to have a higher C-score (i.e., better performance if held out from training). The work we cited actually argues the following. Hardt et al. (2016) present theoretical results that bound the distance between trained weights and random initialization by the number of SGD steps, and show that the generalization gap is small if training completes in relatively few steps. Toneva et al. (2019) study learning speed--and in particular, whether examples are forgotten each epoch--and they informally and anecdotally relate learning speed to notions of outliers and mislabeled examples, not to generalization per se. Mangalam & Prabhu (2019) show that examples learned first are those that could be learned by shallower nets, not that the examples generalize better.
>
> **Ablation study**
> We thank the reviewer for this important suggestion. To ascertain the sufficiency of 2000 model samples in obtaining a reliable estimate of the C-score, we partitioned the 2000 models on CIFAR-10 into two disjoint groups of 1000 and computed C-scores for each group. The Spearman rho correlation is larger than 0.99 between the two estimates as well as between any one of the 1000-model based estimates and the 2000-model based estimate. This shows that 2000 is adequate to produce stable rankings.
>
> **why s=70 for ImageNet?**
> We choose s=70 for ImageNet based on the observations from Fig.6a: we find that the optimal s for MNIST, CIFAR-10 and CIFAR-100 increases as the complexity of the dataset increases. Because ImageNet is more challenging than CIFAR-100, so we chose s=70 rather than s=50 (optimal for CIFAR-100).
>
> **Code can be made public anonymously**
> Thanks for the suggestion. We shared our code anonymously at https://gofile.io/d/KaPxsg

---

> > ### Comment · AnonReviewer2 · 2020-11-21
> > **Thanks for the response!**
> >
> > Based on the additional ablation study, does this mean using 1000 model samples when estimating C-score is enough and 2000 is an overkill? I wonder what if you use even smaller number of model samples to estimate C-score? Essentially, I want to see the sensitivity of C-score with respect to the number of model samples used.
> >
> > As for the practical implication, I wonder how good is C-score on finding out-of-distribution / mislabeled samples when comparing with existing methods?

---

> > > ### Author Response · Authors · 2020-11-24
> > > **Thanks for the suggestion!**
> > >
> > > Thank you for suggesting that we investigate the minimum number of models needed to obtain a replicable estimate of the C-score. We conducted this investigation (Appendix F in the updated manuscript) and obtained an interesting result. We compared the integral C-score based on 1000 models to the integral C-score based on _m_ models, and we found a good correlation for as few as (roughly) _m_=64 models. However, the integral C-score requires training models for various subset ratios (9 different subset ratios in our simulations), so the total number of models needed is roughly 64 X 9. If we want to obtain a reliable estimate of the C-score under a single fixed subset ratio, we find -- for _s_ in [30, 60] -- we need 512 models in order to get a > .95 correlation with the score obtained from the 1000-model hold-out set.  So it appears that whether we are computing the integral C-score or the C-score for a particular subset ratio, we need to train on the order of 500-600 models.

---

> > > ### Author Response · Authors · 2020-11-24
> > > **Outlier detection**
> > >
> > > Our examination of outlier detection was directed at a comparison with an alternative measure, forgetting events, which was shown to be a strong indicator of hard examples during neural network training and capable of outlier detection [Toneva et al., 2019, Figure 3]. We find that the C-score is a more reliable discriminator of outliers than is the forgetting-event measure (Figure 9c).  Nonetheless, we acknowledge the reviewer's concern that we have yet to evaluate the C-score against more traditional outlier detection methods.

---

### Official Review · AnonReviewer1 · 2020-10-27
**Interesting paper**

**Rating:** 5
**Confidence:** 4

**Review:**

This paper proposes yet another score, the consistency score, to evaluate the importance of individual data samples built on held-out performance. This time the authors focus on a dynamic training perspective by studying the effect of different sample sizes. The paper is generally well written and easy to follow, despite some long confusing lines. The experiments are quite comprehensive, equipped with proper discussions. In particular, this work offers an insightful characterization of the better generalization ability of SGD through sample-level stratification according to C-score.

Nevertheless, I still have a few concerns about the motivation and the novelty of the paper:
1. Lack of theoretical or empirical justification of the proposed metric. As motivated by the authors, consistency score is designed to tell regular examples from exception examples, measured by the held-out performance. But it seems more reasonable to use the relative accuracy (standard acc - held-out acc), i.e., the influence value (Feldman and Zhang 2020), to measure the learning difficulty of each individual example. Perhaps the authors should put more discussions on this topic and show the difference between the two metrics. Similarly, the authors also lack a discussion of the necessity of designing such a new metric based on held-out acc, which is hard to compute. As pointed out in Figure 9, there is a high correlation between C-score and accuracy (or prediction entropy), which also seems a good criterion for evaluating the quality of an individual example and is also quite easy to compute. Hence, an involved discussion/comparison of the choice of the particular formulation is needed, either theoretically or empirically. Without this, the novelty and motivation of this work would be questionable.
2. Despite the inspiring introduction, the main part of the paper lacks a clear storyline and seems verbose sometimes. For example, Figure 1,3,4,5 basically tell the same natural story (C-score proportional to sample complexity) without much additional insight on the problem. The authors should focus on discussing the interesting phenomena indicated by the new metric, rather than listing every possible experiment result. As for as I can tell, a major difference/contribution of this work is that it involves the dependence on sample size, which allows it to study the dynamic training behavior. I think the authors should focus on this perspective. For example, I do not see the necessity of discussing the average C-score of different sample sizes. Except for computational stability, I wonder whether this metric really means something, not to mention that the authors do it in a rather heuristic way, by assuming a uniform distribution over sliced sample sizes. Also, Figure 6 seems to deal with a technical issue with limited computation (again a trivial phenomenon), and it could be deferred to the appendix.
3. Personally, I find two insightful phenomena in this work worth more involved discussion. First, Figure 2 demonstrates quite different behaviors of examples with different C-scores, that the bad ones can get even worse with a large subset! Second, Figure 8 shows the C-score helps understand the superiority of SGD from a (not only intuitively understood but also empirically shown) sample-level perspective. If the authors could stick to these two stories and dig deeper, I believe the paper would be more impressive.

In all, I find this paper is well written with thorough experiments around the proposed metric. However, as mentioned above, it would be a regret that the paper lacks more in-depth discussion. I would be pleased to upgrade the score with positive feedbacks from the authors.


Minor points:
1. The interplay between generalization and overparameterization is not discussed at all in this paper, so the authors had better not include it in the title.


Reference:
Feldman, V., & Zhang, C. (2020). What neural networks memorize and why: Discovering the long tail via influence estimation. arXiv preprint arXiv:2008.03703.

---

> ### Author Response · Authors · 2020-11-18
> **Thanks for the comments!**
>
> **Lack of theoretical or empirical justification for the metric**
> The metric is essentially the probability of correctly generalizing to a held out instance conditioned on sample complexity. Isn’t this probability at the heart of what we care about in machine learning?  What makes it novel is that it is specific to each example, rather than being an expectation over a data distribution.  To contrast our work with Feldman and Zhang (F&Z): We are interested in the influence of training sets of a varying size on test accuracy for a particular example.  F&Z are interested in the causal influence of including a particular example in the training set on test accuracy for another example.
>
> For F&Z’s influence score, one can compute the influence of an example on itself, which F&Z call the memorization score, and indeed the memorization score is anticorrelated with our C-score. However, there is a dissociation: the case in which the self-influence score and the C-score are both low. This case corresponds to an example that is not memorized by the training procedure and is dissimilar from other examples in the training set.
>
> **High correlation between C-score and accuracy**
> This correlation is a key finding of our paper. Indeed, the C-score is an expensive quantity to estimate. As stated in the abstract and throughout the paper, our quest was to identify efficient proxies to the C-score.  We are happy that the reviewer appreciates that the learning speed (integrated accuracy or confidence over training) is “a good criterion for evaluating the quality of an individual example and is also quite easy to compute.”
>
> **Why compute an average C-score over subset ratios**
> We are interested in ranking the examples according to their consistency. We explored the range of subset ratios to determine the interaction of the subset ratio on the ranking. Fortunately, the rankings are almost identical across subset ratios, except that at the low and high subset ratios, there are floor and ceiling effects, respectively, that make rank discrimination less reliable. To minimize variance of our estimates, we chose to aggregate across subset ratios. We use uniform weighted aggregation, which is a standard numerical technique for estimating the integrated area under the curve, but other aggregation measures would yield similar outcomes.
>
> **Deeper discussion of insightful phenomena.**
> We are pleased the reviewer noted these intriguing phenomena and indeed we should investigate both. However, our focus in this article was to formulate the analytical tool for investigating the structure in data sets. To have a useful tool, we first need to establish a computationally tractable proxy, which was where our investigations led. Concerning the systematic decrease in C-score for larger subset ratios (Fig 2), we are inclined to believe that this phenomenon is related to error that children make in learning verb past tenses (see introduction) where exception verbs are first learned but later in the course of development are over-regularized.

---

### Official Review · AnonReviewer4 · 2020-10-28
**Some problems**

**Rating:** 4
**Confidence:** 4

**Review:**

This paper proposes a consistency score (C-score) that measures the expected accuracy of a held-out instance averaged over different training sample sizes, which can be useful for analyzing learning dynamics and generalization performance.

I see some problems with the proposed approach. First of all, the proposed metric is computationally very expensive since 2,000 models need to be trained for each s. Can't we simply use the learning speed of each test sample to achieve similar goals? Measuring learning speed should be computationally much simpler since it does not require training 2,000 models if we measure the learning speed for test or validation samples. Those test samples are automatically held out since they are not in the training dataset. In fact, authors show strong correlations between C-score and other measures based on learning speed in Fig. 9. Are there *real* benefits of using C-score over simpler alternatives such as measures based on learning speed, e.g., better outlier detection performance, better accuracy for detecting mis-labeled samples, better analysis of learning dynamics and generalization performance, etc.? If there's no such *real* benefit, then it's hard to justify the use of C-score considering its high computational cost.

At s=70,80,90%, there will be a lot of overlap of samples among 2,000 subsets and I don't see much value in distinguishing such cases (70%, 80%, 90%) with such fine granularity. In Fig. 4(a), they (70%, 80%, 90%) do not show much different anyway while there's a huge difference in the histograms between 10% and the rest (20% ~ 90%). To solve this problem, I suggest using something like s=1%, 2%, 4%, 8%, ..., 64% (log scale) instead of 10%, 20%, ..., 90% (linear scale). This way, we can also see the effect of very small n, which I believe is an interesting regime to study. Using log scale may also be better for comparing consistency profiles for different datasets such as CIFAR-10 and CIFAR-100. Fig. 2 shows the consistency profile curves start at around 0.2 for CIFAR-10 at s=10% while they are more spread from 0.0 to 0.85 for CIFAR-100 at s=10%. This may be an artifact of having coarse-grained intervals for s. By using log scale, we may be able to see a similar behavior between CIFAR-10 and CIFAR-100 only shifted in log scale, e.g., the consistency profile curves for CIFAR-100 may also start around 0.2 when s=1%.

I am not sure how to justify the definition of C-score that simply averages the consistency profile over uniform s from 10% to 90%. Why uniform instead of non-uniform? Why consider averaging instead of max, min, etc.? There are many other possibilities. Taking an average over uniform s seems ad hoc to me. Can authors provide a good justification? If there's no good justification, it may make sense to define C-score at a particular value of s (what authors call a point estimate) without even defining C-score averaged over s. But, this issue does not seem crucial.

How about using a higher learning rate for samples with low C-scores to facilitate learning from small number of irregular samples? This seems to be the opposite of what authors are suggesting in Figs. 8 and 9.

It would be great if authors can improve the performance of optimizers based on the observations made in Figs. 8 and 9.

---

> ### Author Response · Authors · 2020-11-17
> **Thanks for the comments**
>
> **"Can we simply use the learning speed of each test sample?"**
> We are confused by the reviewer’s comment.  The reviewer is essentially reiterating one of the key contributions of our work. To elaborate, we proposed the C-score as a theoretical measure related to an example’s generalization performance. We estimated the C-score with 2,000 expensive models. But the focus of our work, as stated in the abstract and throughout the paper, is to identify efficient proxies to the C-score. The learning speed of an example is the proxy we found best (Section 6).  As a result of our efforts, future researchers can use learning speed (of a training example) for the purposes that we and the reviewer have listed.
>
> Did the reviewer intend to say “test example”, as opposed to the learning speed of a training example? The issue with examining learning of a held-out or test example is that if it has a low C-score, it is unlikely to be learned on any one training run. In contrast, because we are studying overparameterized models, most training examples will be learned, allowing us to assess their relative learning speed.
>
> **Log scale subset ratios**
> We appreciate the reviewer’s suggestion of examining subset ratios over a log-linear range. And we certainly will do so for future data sets. However, we note that for the data sets we studied, the subset ratios we examine are sufficient for characterizing the ranking of C-scores in that the low subset ratios provide good discrimination of the ‘easy’ examples and the high subset ratios provide good discrimination of the ‘hard’ examples, and the curves in Figure 2 are sufficiently smooth that we can grasp their shape.
>
> We also noted the interesting observation that the CIFAR-100 profiles are more spread out than CIFAR-10 at low s (Fig. 2). We have the following intuition about what’s happening: in Fig. 3 we plotted the top ranked examples in the two datasets. It turns out that, _among the most easy examples_, CIFAR-10 has the two classes ‘automobile’ and ‘truck’ that are visually mixing with each other. So those examples did not get a very high score when s is small. On the other hand, on CIFAR-100, the high ranked examples are highly structured _and_ not similar to instances from different classes, so they are easy to learn even with small s.
>
> **Justification of uniform average over s**
> The goal of aggregating C-scores across subset ratios is to characterize each curve with a single scalar. Fortunately, because of the consistency of the rankings across subset ratios, the exact form of aggregation does not have much impact on the ranking of the aggregated scores. We chose to average, which is a standard numerical technique for estimating the integrated area under the curve, but other aggregation measures would yield similar outcomes.
>
> **Higher learning rate for low C-score examples to encourage learning from a small number of irregular samples**
> Thanks for the interesting suggestion! We want to emphasize that this is exactly why we think having a clean formulation and efficient proxies for a consistency score that quantize the structures of the training set is an important contribution: it opens up possibilities to do fine grained studies of learning curricula and analysis of learning behaviors. Concerning the  reviewer’s specific suggestion,  our results in Figs. 8 and 9 indicate that a _lower_ learning rate for low C-score examples benefits at least the early stages of training. The intuition is that over-parameterized models are implicitly regularized when learning on a relatively large collection of mutually consistent instances with variations. However, we note that curriculum learning is a very active research topic in deep learning and so far we have very limited understanding of what is the best curriculum and why. We believe our paper adds important analytical tools to the study of curriculum learning.

---

### Decision · Program_Chairs · 2021-01-07
**Final Decision**

**Decision:**

Reject

**Comment:**

This paper proposes the c-score, which is the aggregation of a "consistency profile" that measures per-instance generalization.  Naive computation of the c-score is expensive and thus requires an approximation.  The paper then uses the c-score to analyze several image benchmarks and their learning dynamics.

While the reviewers found the experiments to be well-done, their primary concern was over the novelty and ultimate usefulness of the c-score.  As R1 and R4 point out, the c-score correlates with other known measures such as accuracy and training speed.  The authors claim this is a contribution.  In turn, it is hard to tell if the c-score is a true metric of interest or a recapitulation of what is already known.  No reviewer was in favor of acceptance.